# “Let Food Be Thy Medicine”: Gluten and Potential Role in Neurodegeneration

**DOI:** 10.3390/cells10040756

**Published:** 2021-03-30

**Authors:** Aaron Lerner, Carina Benzvi

**Affiliations:** Chaim Sheba Medical Center, The Zabludowicz Research Center for Autoimmune Diseases, Tel Hashomer 5262000, Israel; carina.ben.zvi@gmail.com

**Keywords:** gluten, nutrients, intestine, brain, neurodegeneration, gut-brain axis, cross-reactivity, sequence homology, BLAST

## Abstract

Wheat is a most favored staple food worldwide and its major protein is gluten. It is involved in several gluten dependent diseases and lately was suggested to play a role in non-celiac autoimmune diseases. Its involvement in neurodegenerative conditions was recently suggested but no cause-and-effect relationship were established. The present narrative review expands on various aspects of the gluten-gut-brain axes events, mechanisms and pathways that connect wheat and gluten consumption to neurodegenerative disease. Gluten induced dysbiosis, increased intestinal permeabillity, enteric and systemic side effects, cross-reactive antibodies, and the sequence of homologies between brain antigens and gluten are highlighted. This combination may suggest molecular mimicry, alluding to some autoimmune aspects between gluten and neurodegenerative disease. The proverb of Hippocrates coined in 400 BC, “let food be thy medicine,” is critically discussed in the frame of gluten and potential neurodegeneration evolvement.

## 1. Introduction

The gut–brain axes connote a very complex and a challenging topic that tries to decipher the cross-talks between the two extrema, hence functionally dependent compartments. For decades, the brain dominated the arena. However, the increased knowledge on gut performances, mucosal and luminal eco-events, and immune surveyance and regulation have flipped the dogma [1]. It appears that one can’t without the other. Facing the environment, the primary immune function of the intestine is to induce tolerance and to negate the non-self for a long-term homeostasis.

Neurodegenerative diseases are characterized by the progressive loss of structure or function of neurons, finally resulting in their death. The most frequent ones are Parkinson’s (PD) and Alzheimer’s diseases (AD). They are genetically mediated but, the role of environmental factors is constantly unraveled. More specifically, the place of the nutrients, dysbiome and its metabolome, luminal enzymatic modification of naïve proteins, increased permeability and the resulting leaky gut is gaining knowledge [2,3]. In parallel, brain–gluten cross-reactive antibodies and peptides’ sequences identity between gliadin peptides and cerebral antigens are constantly reported. Hence, strengthening the autoimmune processes of molecular mimicry in neurodegenerative conditions [4]. In this regard, the present narrative review describes the potential detrimental effects of gluten ingestion on neurodegenerative disease evolvement.

The first part of this review addresses the relationship between gluten and neurodegenerative diseases, while the second part screen the cross-reactivity and the sequence homology between gluten peptides and human central nervous systems’ antigens. The literature search covered the period 2000–2020 and included studies that describe gluten/gliadin association with neurodegenerative disorders. Research studies, reviews, and case–control series were included, while case reports were excluded. The literature search was performed using the PubMed, MEDLINE, Embase, Scopus, and Cochrane Database of Systematic Reviews databases to identify the most relevant information. The following search keywords were used “gluten” or “gliadin” AND “Neurodegenerative” or “neuroinflammatory” or “neuropsychiatric” AND “Alzheimer disease” or “Parkinson’s disease” or Amyotrophic Lateral Sclerosis, and Multiple Sclerosis, were searched. Additional studies were identified by examining the reference list of the retrieved articles. The search was limited to articles published in English. Relevant articles were selected for full-text review on the basis of screened titles and abstracts. Since primary and personal data were not included, human rights approval was not necessary. Sequence homologies between related human brain antigens and Gluten/Gliadin peptides were explored. The UniProt Knowledgebase (www.uniprot.org/, accessed on 15 December 2020) was used to extract α/β-Gliadin MM1, (UniProt: P18573). The NIH/US National Library of Medicine’s Basic Local Alignment Search Tool (BLAST) sequence matching program, (blast.ncbi.nlm.nih.gov/Blast.cgi, accessed on 24 December 2020) was used to identify sequence homology between Gliadin epitopes and central nervous system (CNS) antigenic sequences. The Immune Epitope Database (www.iedb.org, accessed on 21 March 2021) was searched to extract all human antigens epitopes that are implicated in central neuronal diseases. This included Alzheimer disease, Parkinson’s Disease, Amyotrophic Lateral Sclerosis, and Multiple Sclerosis. The aggregated epitopes were “Linear Epitopes” of “B cells” OR “HLA I” OR “HLA II” AND were rated as “Positive Assays”. In addition, neuronal epitopes that were found in the literature search to have cross reactivity or sequence homology with Gliadin, were included in this epitopes list [5,6,7,8,9,10,11]. Following this, a pairwise local alignment tool was used, EMBOSS Matcher (www.ebi.ac.uk/Tools/psa/emboss_matcher/, accessed on 4 October 2019). This tool implements an algorithm that is based on the Bill Pearson’s Lalign application, version 2.0u4 (February 1996). Using a Python script, the EMBOSS Matcher was executed on each of the neuronal epitopes against the Gliadin sequence and the following cutoff parameters were used to express the results: peptide length ≥ amino acids, similarity ≥50% and identity ≥50%.

## 2. Gluten and Tissue Transglutaminase Potential Involvement in Neurodegeneration

The world market of wheat surpasses all other crops combined and gluten is its major protein, comprising 80% of the wheat’s proteins. It is the most favored staple food worldwide and a major food additive in the processed food industries [12]. Gluten essentiality, its protein’s quality and consumption necessity for human health is debatable. However, there is no doubt regarding its inductive role in gluten dependent disease like celiac disease (CD), dermatitis herpetiformis, gluten ataxia, gluten allergy, and potentially in non-celiac wheat sensitivity [13,14].

The two most frequent conditions are non-celiac wheat sensitivity and CD with incidences of 1–6% and 1–1.5%, respectively. Dermatitis herpetiformis, gluten allergy and gluten ataxia are much less frequent with an prevalence of 1:10,000, 0.2–0.5%, and very rare, respectively [15].

Its potential role in extraintestinal manifestation of CD and remote organs’ pathologies is well reported [16,17]. In addition, its place in the enteric eco-events in the trajectory of the gut-brain axis was recently described [18]. After ingestion, gluten is digested by luminal proteases to various gliadin peptides which are the offending molecules in the gluten-related diseases [19]

Most recently, gluten’s potential role in neurodegeneration was suggested by Mohan et al. [20]. The authors comprehensibly described and suggested that dysregulated microbiome, anti-tissue transglutaminase (tTG) 6 in celiac ataxia, various gut derived biomolecular condensates or extracellular microbial vesicles might play parts in neurodegenerative evolvement. Furthermore, the authors alluded to various therapeutical strategies to prevent or treat those brain conditions. Not only gluten withdrawal, but probiotics, and some nutraceuticals, such as phyto and synthetic cannabinoids, were suggested to mitigate dietary gluten-induced neurodegeneration [20]. The clinical basis for gluten associated CNS diseases comes from the numerous neurological, psychiatric and behavioral manifestations, extensively describes in gluten induced conditions in Table 1 [16,21,22,23,24,25,26,27,28,29,30].

Substantial reinforcements for the gluten-brain axis are coming from epidemiological, biochemical, pathophysiological and nutritional scientific sources. Epidemiologically, gluten consumption, gluten dependent diseases and neurodegenerative/neuroinflammatory diseases’ incidences are rising, at least in the last decades [12,13,31,32,33,34]. Intriguingly, the autoantigen of CD, the tTG, also called TG2, is a pleiotropic enzyme expressed ubiquitously and abundantly, in all tissues in the body, including the human cerebrospinal fluid and brain [35,36]. Interestingly, it can occupy a cytoplasmic, trans membranous or extracellular position and recently specific inhibitors were suggested as a new therapeutic strategy to treat neurodegenerative conditions [37]. Gluten/gliadins, being tTG preferred substrate can be deamidated or transamidated [38,39]. This posttranslational modification of the naïve proteins, turns them to immunogenic molecules, accompanied by loss of tolerance. Notably, not only tTG is everywhere in the body. Gluten and gliadin peptides, as well, are dispersed extra-luminally to reach the systemic blood circulation [40] and are secreted even in the urine [41]. It is postulated that those urinary gliadin peptides are filtered from the systemic circulation. In many human chronic diseases, the gut is leaking and gluten/gliadin peptides can reach the sub-epithelial compartment in their way to the systemic circulation. Quite recently, trans epithelial transport of gluten was demonstrated, thus facing the local sub-epithelial immune systems [42].

The antibody’s cross reactivity between gliadin peptides and brain/cerebellar proteins is not less interesting and might allude to antigenic mimicry between gluten/gliadin peptides and human brain constituents in brain diseases [43,44]. The gut-brain axes are interesting but not less intriguing with much hidden over the visible [16,18]. Ingested gluten/gliadins are playing part in brain pathologies. Following are several gluten-related detrimental effects and pathways that might impact neurodegenerative conditions. On the systemic aspect, gluten is proinflammatory and proapoptotic and impacts epigenetics [45]. On the gut level, it enhances intestinal permeability by compromising functional tight junction integrity resulting in a leaky gut [12,16,18,45].

Interestingly, increased intestinal permeability was reported, not only in active CD and CD in remission [46], but in the other gluten dependent conditions, namely dermatitis herpetiformis [47], non-celiac gluten sensitivity and non-celiac wheat sensitivity, wheat allergy [48]. When exposed to gliadin/gluten, biopsies from non-CD patients demonstrated a lower, limited, transient zonulin release, paralleled by an enhanced intestinal permeability that never reached the level of increased permeability seen in untreated CD [46,47,48]. Intriguingly, gliadin affects tight junction functional integrity also in normal controls [46].

Gluten consumption affects microbiome/dysbiome composition and diversity [38,45,49]. Reduction of diversity with a significant increase of Proteobacteria and an expansion of *Neisseria*, especially in active adult patients, while treated celiacs showed an intermediate profile between active disease and controls [50]. The ratio between anti-inflammatory bacteria such as Lactobacillus-Bifidobacterium to proinflammatory Bacteroides-Enterobacteriaceae in decreased in untreated CD children [51]. The coresponding metabolome reveald altered levels of free amino acids and volatile organic components. To our knowledge, the effects of gluten/gliadin on the intestinal microbiota, its metabolomic effects and the gut-brain relationships in neurodegenerative conditions is still lacking. Down to the cellular level, it decreases viability and cell differentiation, induces apoptosis, and suppresses DNA, RNA, and glycoproteins synthesis [45]. Gluten has substantial effects on the immune performances. It increases neutrophils migration, Th17 cells activity, NKG2D expression and TLR signaling pathway. It affects adaptive and innate immune systems and T reg cells functions [45]. However, a word of caution is advised since most of those side effects were describe in vitro and on animal models and there are not enough studies performed in human [45].There are no defined recommendations to start gluten-free diet (GFD) in non-celiac gluten dependent disease, nor in other autoimmune diseases or neurodegenerative condition, unless gluten related diseases are associated and properly diagnosed [45,52,53]. On the other hand, before implementing gluten withdrawal, one has to consider the difficulties in compliance and the various gluten free-diet side effects [54,55].

## 3. Potential Involvement of Tissue and Microbial Transglutaminase in Neurodegeneration

A total new aspect that might apply to the present topic is the bacterial member of transglutaminase family, namely the microbial transglutaminase. Despite having low sequence homology, its functional imitation of the tTG, to modify gluten/gliadin peptides, is similar [56]. Both avidly deamidate or transamidate those molecules [38,39]. Being a survival factor that is secreted by various prokaryotes including the luminal microbiome it is widely used as a process food additive. Its enzyme gluten modifying capacity in the food industries and intestinal compartment was extensively reported [56,57,58,59,60]. More so, most recently the microbial transglutaminase was suggested as a new environmental factor for CD induction [59,60,61]. Since the microbial and the tTGs share similar enzymatic activity on gluten and brain tTG is involved in brain diseases and since trans epithelial transport of microbial transglutaminase was recently demonstrated [42], not surprising is that the bacterial transglutaminase was hypothesized to be involved in human neurodegeneration [62].

## 4. Gut–Brain Axes

The pathways connecting the gut luminal eco-events with the brain and the mechanisms by which ingested nutrients protect or induce neuroinflammatory/neurodegenerative diseases are far from being understood. Blood and lymphatic vessels can carry gut originated immune cells, antibodies, immunogenic proteins, immune complexes, cytokine and lymphokines. The local immune system can transfer systemically autoantibodies and proinflammatory cytokines. The enteric nervous systems are connected to the brain via the vagal nerve and the para-spinal neuronal routs. In fact, multiple epithelial (enterocytes, enteroendocrine) and sub-epithelial (dendritic and entero-glial) interconnected sensing cells are surveying the luminal events and can mediate the information upwards to the cephalic compartment [16,17,18]. However, there is still plenty to explore in this Pandora enigmatic puzzle.

## 5. Gluten Withdrawal in Neurodegenerative, Neuropsychiatric and Brain Autoimmune Diseases

As a preliminary proof of concept, GFD was reported to be beneficial in some of those neurodegenerative, neuroinflammatory, autoimmune and neuropsychiatric brain diseases (Table 2). However, in some of Table 2 cited references the patients were additionally diagnosed with CD and it is difficult to distinguish between the differential effects of GFD through CD or directly on the brain. Secondly, it should be stressed that gluten restriction is an adjuvant therapy in those brain conditions and doesn’t represent the main or the exclusive therapy.

## 6. Cross-Reactivity between Wheat/Gluten and Brain Tissue Components

Cross-reactive antibodies between self-components and environmental epitopes are a well described biologic phenomenon, as schematically illustrated in Figure 1. In this regards, molecular mimicry between anti-wheat/gluten and brain tissue component were shown by cross-reactive antibodies [8,9,10,78,79]. For example, when autistic children were checked, an eight amino acids sequence similarity of gliadin and cerebellar neural tissue was detected [8]. The authors suggested that cross-reactive antibodies against gliadin peptides and Purkinje cells might play a role in some neurological manifestations in childhood autism. More so, when anti-food antibody derived from six common foods were checked against 65 different tissue antigens, anti-wheat antibody cross-reacted with 15 tissue antigens. Concentrating on brain components, the anti-wheat antibodies reacted against dopamine receptor, neurotropin, alpha enolase and not surprisingly also against tTG which is the autoantigen of CD [10,79]. Interestingly, anti-gliadin antibodies also reacted against glutamic acid decarboxylase (GAD65), an enzyme involved in the production of γ-aminobutyric acid (GABA), which is a prime inhibitory neurotransmitter that, when dysregulated, is implicated in both depression and anxiety [65,78].

The cross-reactivity of anti-gluten/-gliadin/-wheat antibodies with cerebral molecules father attest for the gut-brain axes connecting those heavily consumed nutrients to chronic brain conditions. Being more updated and contemporaneous, the current SARS-Cov-2 was recently defined as an auto immunogenic virus [80,81,82]. In fact, 17 autoimmune diseases and 13 various autoantibodies associated with COVID-19 infection, were reported and the list is continuously expanding [80]. Zooming on the brain, multiple immunogenic epitopes of SARS-CoV-2 have high degree of homology with human brain proteins [83]. Most recently, screening 55 different human tissue antigens, Vojdani A et al., reported on SARS-CoV-2 cross-reactivity with brain tissue antigens like myelin basic protein, neurofilament protein, amyloid-beta, alpha-synuclein, synapsin, GAD65 and tTG-6 [84]. Antibodies against those neural protein targets are depicted in patients with neuroautoimmune and neurodegenerative conditions such as AD, PD, multiple sclerosis (MS), and ataxia [85,86]. Indeed, the current covid-19 pandemic put the PD patients at increased risk for deterioration, worsening their motor as well as non-motor symptoms [87]. The same holds for AD with negative impact on the patients’ cognitive and psychiatric functions [88].

Intriguingly, cross-reactivity between amyloid-Beta 1-42 and tTG and microbial transglutaminase were observed [86]. The author suggested that those cross-reactive antibodies may contribute to intraneuronal deposition of A-Beta-P-42 in AD. It should be stressed that both enzymes, the human tTG and the microbial transglutaminase were described as potential drivers of systemic autoimmunity and gluten is a prime substrate for both of the enzymes [38,39,62]. Finally, most recently, sequence homology between wheat and tTG and alpha synuclein was observed (Vojdani Aristo, personal communication), suggesting molecular mimicry between nutrients, self-tissue antigens and alpha synuclein in PD development.

Taken together, environmental factors, be it nutrients like gluten or infections like the covid-19, might impact neurodegenerative disorders. Cross-reactive antibodies might play a role in neuroinflammatory, neuropsychiatric or neurodegenerative condition. Cross-reactive antibodies allude for autoimmune molecular mimicry, but antigenic sequence identity might strengthen the gliadin-degenerative brain relationship.

## 7. Sequence Homology between Gluten and Brain Tissue Components

Encouraged by the cross reactivity between gluten epitopes and human brain antigens, a Blast search was conducted to identify significant peptides’ sequences identity/similarity between both components. The Glutamate Receptor Ionotropic NMDA-Associated Protein 1 (GRINA) belongs to the Lifeguard family and is involved in calcium homeostasis [89]. This Protein Lifeguard 1, (UniProt: Q7Z429) was identified to have high sequence homology with Gliadin epitopes at identity level of 85.7% (Table 3). Based on this homology, Gliadin peptides may interact with GRINA and interfere with its functionality, which is relevant in many of the extraintestinal manifestations, such as schizophrenia [6]. In a recent study, about one third of the patients with schizophrenia harbor elevated inflammation and IgG antibodies against gliadin (anti-AGA). They showed an increased gut permeability and higher levels of anti-GRINA antibodies that were associated to anti-AGA levels [90].

As summarized in Table 3, multiple human brain antigens share sequence homology with gluten epitopes. Using the Immune Epitope Database and the EMBOSS Matcher tool, the number of proteins that had sequence homology at identity level ≥50% with Gliadin epitopes came down to 17, and the number of identified epitopes was 52. These proteins could be divided into the following categories.

## 8. Neurodegenerative

Synapsin I is a major immunoreactive protein where 17 shared epitopes were identified. Synapsin is a neuron-specific cytosolic phosphoprotein, present in most of the nerve terminals and coats synaptic vesicles. It binds to the cytoskeleton, and is believed to function in the regulation of neurotransmitter release. It affects nitric-oxide functions at a presynaptic level. Anti-AGA from patients with CD identified epitopes from this protein. Although, the potential pathogenic role of SYN1-cross-reactive anti-gliadin antibodies is still unclear [7,9], increasing evidence substantiated the relevance of alterations in synapsins as a major determinant in many neurological disorders, including AD, MS, bipolar disorder, psychosis, schizophrenia, depressive disorder, Huntington’s disease, amyotrophic lateral sclerosis, autism and epilepsy, as demonstrated by both genetic and functional studies [91,92].Amyloid-beta precursor protein is a key protein in AD and is considered one of the main components and inducers of the built-up plaque in the brain. Autoantibodies against this protein are detected in patients with AD and have been shown to cause neuronal degeneration in individuals with compromised blood brain barrier [10,85,93,106].Alpha-synuclein is a neuronal protein that plays several roles in synaptic activity such as regulation of synaptic vesicle trafficking and subsequent neurotransmitter release. This protein is a pathogenic hallmark of PD, and related to dementia with Lewy bodies, and multiple system atrophy [94].Amyloid beta A4 precursor protein-binding family B member 1 (APBB1IP) is another protein with a significant homology of 15 epitopes. It functions in the signal transduction from Ras activation to actin cytoskeletal remodeling. It is associated with Late-onset AD [95].Cerebellar degeneration-related antigen 1: Autoantibodies directed against this protein were found in some patients with paraneoplastic cerebellar degeneration, AD and autism [8].Microtubule-associated protein (Tau) has roles primarily in promoting microtubule assembly in axons, and in maintaining their stability. It is abundant in the CNS neurons, and might be involved in the establishment and maintenance of neuronal polarity. Tau is a key protein involved in many neurodegenerative diseases, including PD and AD [93,96,97,106].

## 9. Autoimmune

Antibodies of the following four myelin proteins were reported in serum of patients suffering from MS [98]:
Myelin-associated glycoprotein is an adhesion molecule that mediates interactions between myelinating cells and neurons.Myelin Oligodendrocyte Glycoprotein Precursor mediates homophilic cell-cell adhesion, and may be involved in maintenance of the myelin sheath and in cell-cell communication.Myelin Proteolipid Protein plays an important role in the formation or maintenance of the multilamellar structure of myelin.Myelin basic protein is the structural constituent of myelin sheath.Myelin-oligodendrocyte glycoprotein is a minor component of the myelin sheath. It may be involved in completion and/or maintenance of the myelin sheath, in cell-cell communication and it mediates homophilic cell-cell adhesion. Antibodies are associated with MS, psychosis, schizophrenia and depressive disorder [98,99].Reticulon-4 Receptor is a receptor for RTN4, OMG and MAG, and sialylated gangliosides GT1b and GM1. Proteomic analysis of cerebrospinal fluid in patient MS was found to contain these proteins significantly dysregulated [98,100].Spectrin alpha chain, non-erythrocytic 1 is a structural protein that ensures vital cellular properties including polarity and cell stabilization. In addition, it is involved in cell adhesion, cell-cell contact, and apoptosis. This protein was also found to be associated with MS [98].Phosphoglycerate Mutase 1 is a glycolytic enzyme that catalyzes step 8 of glycolysis. A proteomics-based analysis revealed high prevalence of autoantibodies against PGAM1 in patients with autoimmune CNS diseases, including MS and neuromyelitis optica [101].

## 10. Neuropsychiatric

Alpha-enolase: This glycolytic enzyme is involved in various processes such as growth control, hypoxia tolerance and allergic responses [107]. It stimulates immunoglobulin production [108] and is a diagnostic marker for many tumors. It is often significantly deregulated in schizophrenic patients compared with controls [102] and might have significance in CD [103].GAD65 is the rate-limiting enzyme for the synthesis of GABA, the major inhibitory neurotransmitter in the CNS. Antibodies against GAD65 are seen in various CNS excitability disorders as well as autoimmune neurological conditions, including stiff-person syndrome, cerebellar ataxia, encephalitis, epilepsy, psychosis, bipolar disorder, depressive disorder, autism, mood dysfunction, anxiety, and behavioral dysfunction [104].

## 11. Discussion

Gluten is the offending nutrient in various gluten-dependent diseases like CD, dermatitis herpetiformis, gluten ataxia, gluten allergy and potentially in non-celiac sensitive conditions [13,14,45]. Despite being the major protein in wheat-the most frequently consumed staple food, it has some harmful effects on human health [13,45,52,53]. It plays a role in the extraintestinal manifestation of CD [16], including in brain pathologies [16,17,18,20,21,24,25,26], hence its involvement in neurodegenerative conditions has just started to be explored. The present review expands on two aspects, namely the cross reactivity and sequence homology between gluten and human brain epitopes, thus reinforcing the molecular mimicry between the two. The classification of the neurodegenerative conditions as ADs is debatable since they don’t fulfill the classical criteria of ADs. PD is an inflammatory condition with some autoimmune aspects [4]. Various autoantibodies against PD associated antigens were described suggesting non secondary, hence primarily causal relationship between the two, resulting in the dopaminergic neuronal loss [4,109,110,111]. Also, AD holds multiple autoimmune features. The vascular-derived anti-neuronal autoantibodies contained in specific brain neurons with degenerative and apoptotic features, including C1q and C5b-9 complement compounds and the permeable blood brain barrier, suggest autoimmunity-induced cell death in AD [112]. More specifically, autoantibodies targeting FcγR-mediated function, tau and ceramide in AD or FcγR-mediated function in PD, were observed to be pathogenic [113]. Most recently, putative 16 autoantibody biomarkers were detected in the cerebrospinal fluid of AD [114] So, not surprising is the above mentions gluten-brain cross-reactivity [8,9,10,78,79] and sequence homology between gluten/gliadin peptides and cephalic epitopes (Table 3). An interesting aspect is the similarity between cephalic antigenes that were detected by both technics, namely, cross-reactivity and sequence homology in relation with wheat/gluten/gliadin components. Neuropsychiatric antibodies against alpha enolase and GAD65 cross-reacted and had sequence similarity with wheat and gliadin, respectively [10,65,78,79]. More so, anti amyloid-beta peptides and anti alpha synuclein antibodies exist in AD and PD [84] and both had sequence homology with gluten/gliadin peptides (Table 3). Anti Purkinje cells and alpha-synuclein antibodies cross-react with gliadin/wheat [8], (Vojdani Aristo, personal communication), alpha synuclein accumulates in Purkinje cells in Lewy body [115] and have sequence homology with gliadin peptides (Table 3). Anti wheat antibodies reacted against dopamin receptors [10,79]. Despite the present lack of sequence homology between gliadin and dopamin receptors, gluten is the major protein in wheat and it is posible that the anti wheat anibodies contain anti gluten antibodies that cross react with the dopamin receptor. Since dopamin loss is a pivotal aspect in PD pathophysiology, one wonders if this cross reactivity might affect dopamin receptors functionality in PD. Based on the above, it is hypothsized that exploring the combined cross reactive antibodies and structure similarity between brain epitopes and gluten/gliadin peptide might shade novel aspects in human neurodegenarative conditions.

An additional interesting pathway by which gluten might affect the brain was summarized by Bressan and Kramer, 2016. It appears that gluten can generate exophines, as shown in animal models [116]. Intriguingly, the gluten originated opioids have higher activity compared to those from casein and many Western societies consume as much as 50 g of gluten, daily [15]. The bad side of those exorphins, when reaching their brain receptors, is their behavioral effects as shown in autism, schizophrenia and psychosis [65]. Decrease social interaction, reduced pain sensitivity, uncontrolled motor activity, disruptive effects on visual and auditory performances were described in mental illnesses as well as in neurodegenerative conditions [65]. The case report illustrating the life-changing amelioration, as achieved by gluten withdrawal in a patient with neuropsychiatric disorder having long term auditory and visual hallucinations, is very intriguing [117]. Finally, some cognitive impairment and “brain fog” associated with gluten-dependent diseases, may respond to gluten elimination [63,118].

If the gluten involvement in neurodegenerative process evolvement is substantiated, a more practical question will be: Will GFD be a novel nutritional therapeutic strategy in preventing or suppressing those conditions? Since adhering to GFD is a tough ally [54,119], several alternative ways to decrease gluten-brain exposure might be envisioned. GF-Mediterranean diet [120], might be very rewarding as Mediterranean diet adherence significantly correlated with 8.4 years of later onset of PD [121]. The dysbiosis in CD and the gluten degrading microbial enzyme (glutenase) capacity might suggest probiotic and lactobacilli enhancing prebiotic therapies [122]. Multiple microbial, fungi and plant proteases were suggested to digest the luminal gluten [123], but it seems that they are not efficient for complete remove of the detrimental gluten peptides, but can help as supplemental therapies. Since one of the drawbacks of GFD is its deficiency in fiber [55] and indigestible polysaccharides are a major nutritional source for the healthy microbiome, prebiotics can boost a normal protective flora. Various functional food supplements, recently summarized by Chander et al., 2018, might decrease gluten exposure. Nutraceuticals are beneficial for PD and AD [93,94,124] and can potentially prevent or treat intestinal barrier dysfunctions and decrease intestinal permeability, thus counteracting the gluten effects on the tight junction functional integrity [12,13,16,45,52,53]. Since tTG and microbial TG can turn naïve gluten peptide to immunogenic molecules [38,39], their specific inhibitors might decrease the cross-linked gluten peptide load on the brain [62,125]. Due to their involvement, transglutaminases were suggested as a therapeutic target for AD [126,127] as well as for PD [128], both aiming to suppress their cross-linked substrate, including gluten peptides, that are rich in proline and glutamine sites [38,39,56,57,58,59,60,61,62]. The harmful effects of industrial processed food, a hallmark of the Western diet, on the gut eco-events and its pro-inflammatory and pro-oxidative, favoring the development of neurodegenerative diseases were recently summarized [129]. Notably, microbial transglutaminase and its gluten peptides preferred substrates are frequently used as food additives [12,38,39]. The resulting cross-linked gluten complexes were recently suggested as potential driver of autoimmunity, not sparing neurodegenerative conditions [56,57,58,59,60,61,62]. Moreover, a trans-enterocytic transport of gliadin and microbial transglutaminase [42] and anti-gluten cross-linked systemic antibodies were recently reported [130,131]. Avoiding process nutrients might decrease the gluten load and diminish its enzymatic cross-linking effects on the human brain. In addition, based on the above, avoiding gluten–brain cross-reactive nutrients and abstaining from gluten–brain sequence similar proteins to minimize molecular mimicry is suggested to be explored in the future.

## 12. Conclusions

It is concluded that Hippocrates was holistically right. Gluten already existed in 400 BC and even much earlier, but sometimes the food is not “thy medicine”, nor the solution. Gluten might be a potential detrimental nutrient in neurodegenerative diseases evolvement. Circulating systemically, being localized in the brain and being a prime substrate for tissue and microbial transglutaminases posttranslational modifications, gluten/gluten peptides should be thoroughly investigated for their pathophysiology in neurodegenerative conditions. The anti-gluten antibodies cross-reactivity and the numerous epitope sequence homologies with CNS peptides direct to the possible pathophysiological pathway of molecular mimicry, operating in neurodegenerative diseases. Figure 2 summarizes the gluten–brain relationship that might operate in neurodegenerative diseases. The quote from Matthew 6:11, 13: “Give us this day our daily bread (…..) but deliver us from evil” might be actual in the frame of gluten induced neurodegeneration and mental illness [65]. The gluten–brain degeneration axis exploration is only in its infancy and deserves extensive research exploration. If substantiated, it could represent a new therapeutic strategy for neurodegenerative conditions.

## Figures and Tables

**Figure 1 cells-10-00756-f001:**
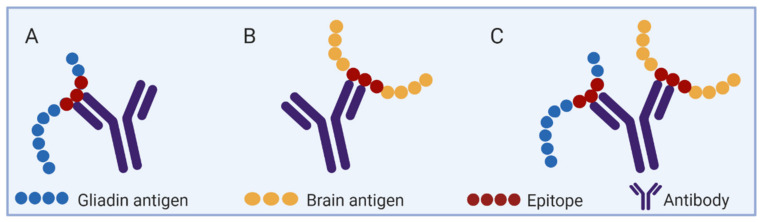
A schematic presentation of cross-reactive antibodies between two separated antigenic determinants. The specific antibody that reacted with each is cross-reacting to both of them. (**A**) Anti-gluten/gliadin antibodies. (**B**) Anti-brain autoantibodies. (**C**) Anti-gluten/gliadin and brain epitopes cross-reactive antibodies.

**Figure 2 cells-10-00756-f002:**
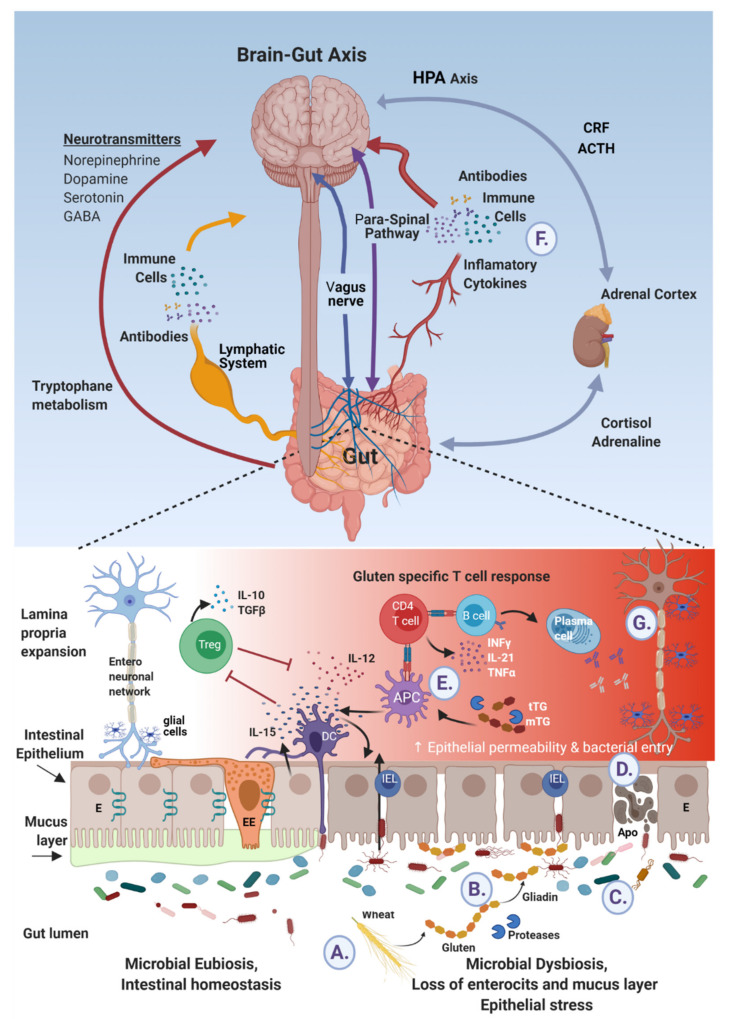
Ingested gluten and gliadin’s peptides cross talks with brain epitopes in neurodegenerative diseases. (**A**) Wheat reach gluten is ingested and digested to gliadin peptides. (**B**) By deamidation and cross-linking, luminal and mucosal tissue and microbial transglutaminases post translate those peptides to immunogenic molecules. (**C**) In parallel, gluten affects the microbiome/dysbiome ratio, resulting in proinflammatory metabolome and harmful microbial constituents. (**D**) This mobilome finds its way, trans- or inter-enterocytically, through the failed tight junction to end up sub-epithelially. (**E**) In addition, the sensing epithelial and subepithelial cells are activated and deliver signals to the adjacent local or systemic blood, lymphatic and neuronal networks (**E**,**F**), respectively. (**G**) Finally brain neuroinflammatory and neurodegenerative processes are affected.

**Table 1 cells-10-00756-t001:** Neuro and psychiatric manifestations in gluten related diseases.

Disease	Neuro/Psychiatric Manifestations	References
Celiac disease	peripheral neuropathy, inflammatory myopathies, myoclonus, myelopathies, headache, migraine, and gluten encephalopathy, epilepsy and seizure disorders, restless legs syndrome. Anxiety, depressive and mood disorders, attention deficit hyperactivity disorder, autism spectrum disorders, schizophrenia	[16,21,24,25,26,27,28,29]
Non-celiac wheat sensitivity	foggy mind’, headache, leg or arm numbness, epilepsy and seizure disorders, gluten ataxia, gluten neuropathy and gluten encephalopathy. depression, anxiety psychosis, schizophrenia, autism, and hallucinations	[27,28,29,30]
Dermatitis herpetiformis	Rarely essential tremor, chorea, migraine.	[22]
Gluten ataxia	Mainly gait and limb ataxia. rarely, myoclonus, palatal tremor, opsoclonus, chorea, Gaze-evoked nystagmus and other ocular marks of cerebellar dysfunction.	[27,28,29]
Gluten allergy	None	[23]

**Table 2 cells-10-00756-t002:** Brain diseases that might benefit gluten withdrawal.

Disease Category	Disease Name	Reference
Neurodegenerative	Alzheimer disease, Cognitive impairment	[63]
	Parkinson’s disease	[20,29,64]
Neuropsychiatric	anxiety	[65]
	depression	[20,65,66]
	“foggy mind”	[15,66]
	schizophrenia	[65]
	autism	[20,65,67]
	psychosis	[20,68]
	bipolar disorder	[65]
Autoimmune	multiple sclerosis	[69]
	autoimmune hypopituitarism	[70]
	gluten ataxia	[20,71]
	autoimmune uveitis	[72,73]
Miscellaneous	migraine	[74]
	chorea,	[75]
	epilepsy	[76]
	headache	[77]

**Table 3 cells-10-00756-t003:** Sequence homology between gluten/gliadin peptides and brain antigen.

Parent Protein	Disease Name	Reference	UniProt	Protein Epitope	Gliadin	L	Sim%	Id%
Protein lifeguard 1		[6,89,90]	Q7Z429	PQGPYPQ	PQQPYPQ	7	85.7	85.7
Synapsin-1	Alzheimer disease multiple sclerosis bipolar disorder psychosis schizophrenia depressive disorder Huntington’s disease ALS Autism epilepsy.	[7,9,91,92]	P17600	PQGPYPQ	PQLPYPQ	7	85.7	85.7
				QAGPPQ	QPFPPQ	6	66.7	66.7
				TTAAAA	TTARIA	6	66.7	66.7
				QRQAGPPQ	QQQPFPPQ	8	75	62.5
				PAPPK	PFPPQ	5	80	60
				ASRVL	SSQVL	5	100	60
				AVKQTTA	AIVATTA	7	71.4	57.1
				QDIASVV	QAIHNVV	7	71.4	57.1
				PPQGGPPQPG	PPQQPYPQPQPF	16	62.5	56.2
				PGPQRQGPP	PGQQQPFPP	9	66.7	55.6
				AVKQTTAAAA	AIVATTARIA	10	60	50
				DVRVQK	DVVLQQ	6	83.3	50
				PPKASGAPPG	PPYCTIAPVG	10	60	50
				RPPPQGGP	QPYPQSQP	8	62.5	50
				TTYPVV	STYQLV	6	83.3	50
				VRSLKP	VPQLQP	6	66.7	50
				VEQAEF	VQQQQF	6	83.3	50
Amyloid-beta precursor protein	Alzheimer disease	[10,85,93]	P05067	LALLAIVAT TARIAVRVP	LALLLLAAW TAR-ALEVP	18	72.2	61.1
				QKLVFFAE	QQLPQFEE	8	62.5	50
Alpha-synuclein	Parkinson’s disease dementia with Lewy bodies multiple system atrophy	[94]	P37840	VVHGV	VVHAI	5	80	60
Amyloid beta A4 precursor protein-binding family B member 1-interacting protein	Late-onset Alzheimer disease	[95]	Q7Z5R6	QATHSV	QAIHNV	6	83.3	66.7
				VPELE	VPQLQ	5	100	60
				STKSL	STYQL	5	60	60
				QFENV	QFEEI	5	80	60
				PAPVP	PQPFP	5	60	60
				LMKAL	LILAL	5	80	60
				IYYGT	IAYGS	5	80	60
				FKNPQ	FQQPQ	5	80	60
				YPELQI	YPQPQL	6	83.3	50
				QKESQY	QSQPQY	6	66.7	50
				QHKMKY	QHSIAY	6	66.7	50
				PELQIERF	PYLQLQPF	8	75	50
				NVVEVL	NVVHAI	6	66.7	50
				LLTQSL	ILQQQL	6	66.7	50
				AGLASR	ARIAVR	6	66.7	50
Cerebellar degeneration-related antigen 1	Alzheimer disease paraneoplastic cerebellar degeneration autism	[8]	P51861	EDVPLLE	EQVPLVQ	7	85.7	57.1
Microtubule-associated protein tau	Alzheimer disease Parkinson’s disease	[93,96,97]	P10636	YSSPGSP	YSQPQQP	7	57.1	57.1
Myelin-associated glycoprotein	multiple sclerosis	[98]	P20916	VSLLC	VQQLC	5	60	60
				PYPKN	PYPQS	5	100	60
				GAWMPS	GSFQPS	6	83.3	50
Myelin Oligodendrocyte Glycoprotein Precursor	multiple sclerosis	[98]	Q16653	EIENL	EIRNL	5	80	80
Myelin Proteolipid Protein	multiple sclerosis	[98]	P60201	TSASIG	TIAPVG	6	66.7	50
Myelin basic protein	multiple sclerosis	[98]	J3QKL5	LCNMY	MCNVY	5	100	60
Myelin-oligodendrocyte glycoprotein	multiple sclerosis psychosis schizophrenia depressive disorder	[98,99]	Q5SSB8	EELRN	EEIRN	5	100	80
Reticulon-4 Receptor	multiple sclerosis	[98,100]	Q9BZR6	GPGLFR	GQGSFQ	6	66.7	50
Spectrin alpha chain, non- erythrocytic 1	multiple sclerosis	[98]	Q13813	SQLLANS	SQVLQQS	7	71.4	57.1
Phosphoglycerate Mutase 1	multiple sclerosis neuromyelitis optica	[101]	P18669	IRHGES	IAYGSS	6	66.7	50
Alpha-enolase	schizophrenia celiac disease	[102,103]	P06733	LVVGLC	LVQQLC	6	66.7	66.7
Glutamic acid decarboxylase GAD65	stiff-person syndrome cerebellar ataxia encephalitis epilepsy psychosis bipolar disorder depressive disorder autism mood dysfunction anxiety behavioral dysfunction	[104,105]	Q9UGI5	MCNVYIPP	VCFWYIPP	8	75	62.5

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
