# Peer review of "“Let Food Be Thy Medicine”: Gluten and Potential Role in Neurodegeneration"

_cells, 2021, doi:10.3390/cells10040756_

Round 1

Reviewer 1 Report

The manuscript is a good entry into the field of gluten and neurodegeneration. The manuscript is well described, figures and Tables are clear and informative.
References are adequate and updated.

I only suggest to include the following comprehensive review related to AD when speaking about amyloid-beta and TAU in line 220 page 9, and line 234 page 10, respectively.

Campora M, Francesconi V, Schenone S, Tasso B, Tonelli M. Journey on Naphthoquinone and Anthraquinone Derivatives: New Insights in Alzheimer's Disease. Pharmaceuticals (Basel). 2021;14(1):33. doi: 10.3390/ph14010033

Reviewer 2 Report

Dear authors,

This is an interesting review trying to describe the connection between gluten and degeneration. I have some questions and comments:

1.Introduction. There are not references on the introduction. Please, include them. 

2. Material and Methods. Since this is a review, you don’t need to include material and methods section. On the introduction , you can refer to the literature research, and the blast analysis.

3.Gluten and tissue transglutaminase section. This section needs to be clarified. It is better explained on the figure. Also, please, describe what gliadin is. 

4. Sequence homology.  I understand the authors did a sequence comparison, however, I would focus only on the proteins that, based on the literature, actually have cross reactivity with gluten/gliadin.

5. Discussion needs to be developed more in deep, based on what you described earlier. 

Thank you very much.

Reviewer 3 Report

As a whole, this review addresses an original aspect of the potential effects of gluten on the brain. However, the rationale is not clear.

In people without intolerance, gluten is not a priori "toxic" to my knowledge. The points made must take into account both the effects of gluten in people with and without gluten intolerance.

This is essential and requires a clear review of the various points in the review, with a clear distinction between sensitive and non-sensitive people.

In distinguishing between sensitive and non-sensitive individuals, it is necessary to clarify:
1) How many people worldwide are affected by gluten disorders, what is the incidence per country, what age groups are affected, what are the consequences on life, what are the treatments. A reminder is necessary.
2) What is the reason for focusing on the effects of gluten on neurological disorders? Are neurological and neuropsychiatric disorders more common in people with gluten sensitivity? Adding a table with precise references is totally necessary.
1) What are the effects of gluten on the microbiota, does it modify the activity of the microbiota in all sensitive individuals more than in non-sensitive individuals, what are its effects on the intestinal barrier in normal and sensitive individuals, which differences? What are the effects on intestinal metabolism: microbiota, intestinal epithelial cells?
2) Are gluten disorders associated with changes in brain metabolism: lipid metabolism (cholesterol and derivatives, neuromediators...).
3) Are there metabolic pathways, in the brain, that are stimulated or inhibited in gluten intolerant people compared to "normal" subjects?
4) The subject as a whole is interesting but needs to be taken up in a context closer to reality based on epidemiological and clinical data. It is essential to add such data to this review in order to make the hypothesis put forward by the authors on the impact of gluten on the brain more credible.

Minor:

1) The title must be modified.

"Let food be thy medicine"*: gluten and potential role in neurodegeneration 

2) The abstract must be rewritten in a more scientific manner.

Round 2

Reviewer 2 Report

Dear Authors,

Thank you very much for addressing all my comments. The manuscript improved a lot and now it is suitable for publication.

Thank you very much.

Reviewer 3 Report

The review has been corrected and gives a global idea mainly on Cross-reactive antibodies and sequence homologies between brain antigens and gluten: this part is very original and well documented. The review is now clear and well presented.

It is a good review that deserves to be published in "Cells".